# Caregiver Representations of Therapeutic Patient Education Programmes for People with Schizophrenia: A Qualitative Study

**DOI:** 10.3390/healthcare10091644

**Published:** 2022-08-29

**Authors:** Corinne Rat, Nicolas Meunier-Beillard, Samuel Moulard, Frédéric Denis

**Affiliations:** 1Clinical Research Unit of la Chartreuse Hospital, 21000 Dijon, France; 2Clinical Investigation Center—EC Inserm 1432, 21000 Dijon, France; 3Delegation for Clinical Research and Innovation, Dijon Bourgogne University Hospital, 21000 Dijon, France; 4Department of Odontology, Tours University Hospital Center, 37000 Tours, France; 5Faculty of Dentistry, Nantes University, 44035 Nantes, France; 6EA 75-05 Education, Ethics, Health, Faculty of Medicine, François-Rabelais University, 37000 Tours, France

**Keywords:** schizophrenia, therapeutic patient education programme, mental healthcare, patient partnership

## Abstract

**Background:** In France, there are two main types of psychosocial educational therapies for people with mental disorders: (1) therapeutic patient education (TPE) or “training”, and (2) psychoeducation. Both types of educational therapy aim to improve disease morbidity, treatment compliance and patient quality of life, but they have very different modes of application. The aim of this study was to interview mental health professionals in order to explore and identify the determinants (barriers and enablers) underlying their acceptance of therapeutic patient education (TPE) in order to facilitate the implementation of these programmes among people with severe mental illness such as schizophrenia. **Methods:** In this multicentre cross-sectional study, we opted for a qualitative approach based on individual semi-structured interviews with 21 mental health professionals trained in TPE, regardless of whether they had practiced it before or not. In accordance with the “Jardé” law (Decree no 2016-1537 dated 16 November 2016 published on 17 November 2016 in the Official Journal of the French Republic). No regulatory approval was required for this study. **Results:** The major themes that emerged from the analysis were grouped into the following conceptual framework: (1) mental health professionals (MHPs) highlight important organizational and institutional challenges that they feel are beyond their scope; (2) MHPs mention in parallel their own perceptions and representations of TPE in the context of mental health care; and (3) MHPs’ representations could hide a lack of knowledge or awareness that would prevent them from appropriating TPE programmes. For each major theme, the sub-themes identified are presented. **Conclusions:** Although TPE is of interest in the process of patient empowerment, we found that caregivers were reluctant to appropriate this approach to care. Efforts must be made in the initial and ongoing training of MHPs to move from a paternalistic model to a patient partnership model, which is made possible by TPE. These efforts must also be firmly supported by health care facilities, and proactive governance is required for the successful implementation of TPE.

## 1. Background

Schizophrenia is a severe and persistent mental disorder that affects 0.7 to 1% of the world population [1] and 600,000 people in France [2]. Schizophrenic patients are exposed to higher mortality and to numerous comorbidities, and their life expectancy is 10 to 15 years lower than the general population (excluding suicide) [3]. The gap in life expectancy between the general population and schizophrenic patients is not acceptable, and all potential means of reducing it must be explored.

In France, there are two main types of psychosocial educational therapies for people with mental disorders: (1) therapeutic patient education (TPE) or “training” and (2) psychoeducation. Both types of educational therapy aim to improve disease morbidity, treatment compliance and patient quality of life, but they have very different modes of application [4].

Psychoeducation aims, usually through group workshops, to inform patients (or their relatives) about their psychiatric disorder and to promote coping skills by providing structured information about the illness and its treatments [5]. Psychoeducation promotes the resolution of emotional, psychological, behavioural and cognitive problems [5]. It has been shown to have a positive impact on patients’ adherence to treatment, autonomy and recovery, as well as being useful for family and friends [6,7]. It is recommended in particular for bipolar disorders and schizophrenia [8,9]. It is considered to be particularly suitable for psychiatry, and, in contrast to TPE, its implementation does not require a legislative framework or compulsory training [8,9]. The evaluation procedures for the programmes developed are under the sole authority of the practitioners and the institutions that offer them. Specific institutional funding for this activity is infrequent and it is mainly part of a project for a single unit or for the psychiatric department in general, or as part of an annual operating grant. It is often carried out on an ongoing basis, through shared staff or redeployment of staff [8,9] (Table 1).

According to the World Health Organisation (WHO), TPE programmes aim to help patients acquire or maintain the skills they need to manage their lives with a chronic dis-ease [12]. The “Hôpital Patients Santé et Territoire” (HPST) law dated 21 July 2009 [10] gave a legislative framework to TPE in France and recognised it as a right for patients, while the decree no 2010-906 dated 2 August 2010 specifies the skills required to deliver TPE programmes as well as the specifications for the authorisation of these programmes by the regional health agencies (ARS) [13]. They provide for a formalised “shared educational assessment”, the definition of a personalised care programme with the patient, the planning of workshops and sessions, and the evaluation of their acquisition and effectiveness [14]. The French National Authority for Health (HAS) has provided a guide to facilitate the annual self-assessment and four-year assessment that are required to continue running the programmes [14]. Table 1 should be moved to this position

Despite the mentioned differences between psychoeducation and TPE, these educational programmes also have similarities in that they help the person to understand their psychological disorders and to acquire self-care skills to manage them in their daily lives. They also provide information in a personalised way, helping to rebuild their identity, develop coping skills and explore the emotions generated by the disorder [8,9,10]. Most TPE programmes are currently built on the basis of the caregivers’ knowledge of the disease. A caregiver in mental health care refers to any person who is involved in the treatment or prevention of the illness or its complications. The doctor, nurse and care assistant can be considered caregivers. Some TPE programmes are delivered in collaboration with a multidisciplinary team and aim to help the patient gain mastery and skills in order to increase their sense of self-efficacy and help them to be more active in their management [15,16]. New approaches even take into account the patient’s experience of the disease and the experiential knowledge gained during their illness [17]. This new role attributed to patients has led to a redefinition of the relationship between caregivers and patients. Thus, the process of empowerment [18] and otherness relationship patterns [19] have been conceptualised to describe an educational position that “allows the subject to exist in his or her health choices and apprehensions in the face of disease”, and also to redefine the caregiver–patient relationship. In these models, patients seem to be at least as well placed as carers to recognize their needs. Thus, TPE works towards changing the practices and attitudes of caregivers towards patients in general. In France, however, TPE is having difficulty gaining momentum in the field of mental health. Only 2.3% of the 1800 TPE programmes listed in France in 2011 were programmes for patients with psychiatric disorders [20]. Although, in 2016, the number of TPEs in psychiatry had doubled since 2011 [21], a search of the ARS websites shows that in 2021 there was a great disparity in the deployment of TPEs in psychiatry. For example, 13 programmes were listed in the Auvergne-Rhône Alpe region, 6 in Bourgogne Franche-Comté and only 4 in the Grand Est region and 2 in Normandy. The care of psychiatric disorders is of course very different from that of other systemic illnesses. Firstly, because health care alone is not enough, and above all, because it alters relationships with others and is a source of stigmatization [22,23]. Moreover, the relationship between the carer and the patient has until now been oriented towards a paternalistic, benevolent and protective position. In this context, the patient’s attempts to decide for him or herself can be perplexing for caregivers because of their representations of how to care for a patient with a mental health disorder [24]. Moreover, beyond the relational dimension of care in psychiatry, the educational approach raises ethical questions and forces caregivers to assess their personal biases about providing TPE to people with psychiatric disorders [22].

Indeed, recent studies on the representations of the main providers of TPE in psychiatry have highlighted that some mental health professionals (MHPs) have a skewed image of TPE, wrongly considering that it is limited to improving treatment compliance [24,25]. The few studies that have looked at the representations of those providing TPE to patients suffering from psychiatric disorders highlight a feeling of lack of exchange and infantilization by some caregivers [25].

Insofar as the effectiveness of a TPE programme is based on the synergy of the patient–caregiver relationship, individual MHP representations must not hinder the support provided to patients in these programmes. It is therefore important to identify the gaps between the expected and actual roles of caregivers, the mechanisms of resistance employed by the individuals involved, and/or the structural and contextual blockages complicating the implementation of health education for schizophrenic patients.

Qualitative research is a social-sciences approach that is based on observation and listening to bring out new non-quantitative data [26]. It seeks to answer the questions of “why?”, “how?” and “what?”, taking subjectivity into account, rather than validating pre-established hypotheses. It may highlight the ambivalence often found in open questions involving the interviewees’ convictions and feelings. Grounded theory in particular is a systematic set of techniques and procedures that enable researchers to identify concepts and build theories or conceptual frameworks from qualitative data. More specifically, grounded theory is focused on psycho-social processes of behaviour and seeks to identify and explain how and why people behave in certain ways, in similar and different contexts [27]. Data generation is aimed at explaining how changes in action–interaction come about in response to different conditions by capturing data in a way that is amenable to identifying and explaining these processes and phenomena. This leads to a better interpretation of the results and to more complex and relevant hypotheses [28].

## 2. Aims

Our aim was to interview MHPs in order to explore and identify the underlying determinants (barriers and enablers) of their acceptance of TPE, in order to facilitate the implementation of these programmes among people with severe mental illness such as schizophrenia, and how changes can be introduced and sustained.

## 3. Methods

In this multicentre cross-sectional study, we opted for a qualitative approach based on individual semi-structured interviews with MHPs trained in TPE, whether they had practiced it or not. The interviewer was a male sociologist trained in qualitative health research who was not involved in the follow-up of patients. A purposive sample was constituted with MHPs from six French psychiatric care centres.

### 3.1. Participant Selection

Interviewees were paramedical and medical caregivers (nurses, nursing staff, nurses’ aides, social workers, pharmacists, and physicians) practicing in psychiatry and trained in TPE. They may or may not have provided a TPE program for patients with schizophrenic disorders in the last 6 months.

The interviews were conducted until the phenomenon of saturation was reached [29]. Theoretical saturation occurs when questioning additional participants does not bring out new information or themes.

### 3.2. Participants

Interviews with 21 participants came from 7 different organisations were conducted between August 2019 and December 2020 (Table 2); 13 (62%) participants had already practiced TPE and 8 (38%) had not. All participants had completed the mandatory 40 h training to conduct a TPE program [14]. Before a TPE program is started, the patient and caregiver choose together the themes to be worked on following an educational diagnosis. They then set goals to be achieved. Individual or group TPE sessions are then carried out or the patient is involved in his or her education. A specific session aims to evaluate the skills acquired and the changes implemented by the patient in his daily life. For the success of this program, the coordination of health professionals involved in the management of chronic disease around and with the patient is essential [11,14]. The characteristics of the 21 participants in the study are presented in Table 2.

### 3.3. Data Analysis

#### 3.3.1. Inclusion Process

In each participating centre, the study investigator identified eligible MHPs. The investigator explained the objectives of the research to each potential participant and mailed them the written information. If the MHP accepted the interview, their email address was given to the sociologist to schedule an appointment for a face-to-face interview.

#### 3.3.2. Interview Guide

The interview guide was based on: (1) the data available in the literature in order to identify the area of focus [21,27]; and (2) the exploratory semi-directive interviews conducted by a health sociologist with a nurse, a nurse manager, a care assistant and a physician corresponding to the study’s inclusion criteria. The semi-structured interview is a qualitative data collection strategy in which the researcher asks informants a series of predetermined but open-ended questions [27].

The following themes were thus addressed: experience, training and professional background, representations of mental health care, representations of and experience with therapeutic education, and needs and expectations for the development of therapeutic education.

#### 3.3.3. Procedure

Due to the COVID-19 pandemic, some interviews were conducted by videoconference (n = 2) or by phone (n = 5), while the others were conducted in person in the mental health centres. Participants met up with the interviewer in a quiet place for 30 to 60 min. The interviewer first reviewed the principles of a qualitative study and reminded the participant that the interview would be audio recorded (anonymously), which was specified in the study information documents provided beforehand. Active listening with interrogative, reiterative or interpretative techniques was used to encourage the professional to explain and clarify their answers and feelings, but without influencing them. A pause was allowed between questions so that participants could recall events and feelings with precision. Notes on non-verbal communication were included in the transcription of interviews to enrich the data.

Interview records were transcribed in their entirety in a text format for later analysis. Data were encoded to guarantee the anonymity of the participants. Starting from the realities of the field and the participants’ discourse, the inductive approach was favoured, in accordance with the requirements of grounded theory [30]. This involves identifying the themes addressed, grouped into major thematic categories (divided into sub-categories), and then proceeding, with a higher level of inference, to a conceptual interpretation of their interactions. The analysis of the interviews proceeded in six distinct main steps:Open coding of the transcribed interviews in order to bring out as many themes as possible from the initial corpus.Categorisation of the codified elements: a careful rereading of the entire corpus to ensure that each category is clearly defined, its properties identified, and the different forms and conditions of appearance of the phenomena specified.Relating the categories: writing more detailed memos and designing explanatory diagrams.Integration of the previous steps in order to identify the essence of the phenomenon.Modelling: the phenomenon, in addition to being described, defined and explained, will then be examined and conceptualised in terms of its dynamics. The structural and functional relationships of each of its constituents were then highlighted.Theorising: careful and exhaustive construction of the “multi-dimensionality” and “multi-causality” of the phenomenon of the relationships between the needs, expectations and representations of the different actors (physicians, nurses, nursing assistants, social workers, and pharmacists).

In order to reduce analysis bias and ensure cross-validation of the data, the interview data were analysed and interpreted first by the sociologist involved in the field. Then, this initial coding framework or set of codes was discussed in an interdisciplinary meeting with the study’s steering committee who could redefine the boundaries of the themes that emerged from the data.

### 3.4. Ethics

After participants were provided with information about the study, their oral consent was collected and they were included in the study. Their contact details were sent to the sociologist in charge of conducting the interviews so that an appointment could be made in accordance with the “Jardé” law (Decree no 2016-1537 dated 16 November 2016 published on 17 November 2016 in the Official Journal of the French Republic). No regulatory approval was required for this study.

## 4. Results

### 4.1. Findings

The major themes that emerged from the analysis were grouped into the following conceptual framework: (1) MHPs highlight important organizational and institutional challenges that they feel are beyond their scope; (2) MHPs mention in parallel their own perceptions and representations of TPE in the context of mental health care; and (3) MHPs’ representations could hide a lack of knowledge or awareness that would prevent them from appropriating TPE programmes. For each major theme, the sub-themes identified are presented.

### 4.2. MHPs Highlight Important Organizational and Institutional Challenges That They Feel Are beyond Their Scope

#### 4.2.1. The TPE Would Be Too Time Consuming

The professionals described a lack of institutional will to provide TPE in their facility due to the lack of time dedicated to this additional activity.

“I think that in fact the difficulty of TPE comes from the fact that it has not been supported by anyone specific...”(Doctors (D)) and (Nursing Manager (NM)).

“They don’t have much time. They are really busy with all the protocols. They have more and more things to do...”.../... “They don’t have much time.../...”(NM).

“And then there’s so much work too, so much nursing and care assistant work, that it’s not easy to detach yourself”(NM).

“We could do it but at the expense of something else.”[Nursing assistant (NA)].

“In the evening, they have finished their day, they want to go home, I understand them, I really don’t blame them, but they want to go home. They have their children too, they have their mother to visit, they have their life. And to commit oneself like that to something very regular is clearly a constraint.” [NA].

In France, psychiatry suffers from a significant shortage of caregivers, which has a major impact on the implementation of new activities.

#### 4.2.2. High Turnover of Professional Staff within the Units

The respondents also presented the issue of high staff turnover because of the difficulty of the work as a factor that was detrimental to the sustainability of the programmes, whether this was due to personal or institutional causes.

“I was running a workshop with a health executive who left the hospital. And so I found myself... a bit on my own with this...”.../... [NA].

“Another obstacle that everyone in the hospital knows, and this was the case for the unit I was talking about, is that in fact all the care workers change departments every 5 years...”.../... [NM and Psychiatrist (P)].

“This is really a big obstacle. Because if there isn’t a team of carers who can come and re-mobilise, re-explain the origin and re-initiate work on these programmes, on these tools, people find it difficult to re-appropriate them and they have to run a programme that was created by others...”[NM and P].

#### 4.2.3. Institutional Communication Needs to Be Improved

The lack of centralisation and institutional organisation also seemed to be lacking within the institutions and accentuated the feeling of having to manage alone and of lacking visibility in the running of the programmes.

“There must have been four or five professionals who really wanted to get involved. We were lucky enough to have the support of the managers, who made it possible for us to make up for the time taken up personally to structure our workshops.”[P and NM].

“When we asked for a bit of material, we had it all, right away. No, we were lucky in that respect, yes.”[NM].

On the other hand, “At the beginning, it was really word of mouth, there would have been no communication if the doctor who was at the initiative of this project had not done a bit of pushing, creating meetings, inviting people, inviting the different partners, we made flyers, brochures, we tried to talk to a lot of people. And it’s true that little by little our colleagues began to understand the interest in better perceiving the indications and why we could propose that, why we could also think like that.” [NM].

### 4.3. MHPs Mention in Parallel Their Own Perceptions and Representations of TPE in the Context of Mental Health Care

#### 4.3.1. TPE and the Emergence of a Conflict of Values

In its approach to care and the new type of relationship with the patient, TPE is contrary to the culture of psychiatric care that carers are used to. The culture around medical care seems to be still strongly anchored in a paternalistic model where the doctor is the decision maker on what the quality and philosophy of life of the patient should be.

“I think that doctors, or even nurses, like to control, to know everything, and not to leave too much autonomy.”[NA and Social Worker [SW)].

Generally speaking, the idea emerged that the initial training of MHPs leaves too little room for the personalisation of work, group work and the development of social skills, which hinders the development of cooperation.

“We are all used to receiving educational content in a rather formatted, vertical way…”, “Really one-sided, yes. And so that’s totally the opposite of what TPE can be, totally…”[D].

“And, in psychiatry, perhaps we are a little behind in this respect, a little behind, because... well yes, clearly diabetes and pneumology have made a lot of progress in all this. And I think that for years, psychiatry has been watching all this progress, that’s it.”[P].

The participants recognize that the organisation of practices is rigid and difficult to change because it calls into question the meaning of “care” in psychiatry.

“When I told colleagues from other departments that were doing therapeutic education in psychiatry, they looked at me wide-eyed when in fact... it’s still shocking, that’s all.”[P].

Even though the general management of the participants in the study had approved the training for TPE, contradictory orders in the field made it difficult to implement these practices and created a conflict of values and a loss of meaning for them.

“Behind all this, it is the importance of the meaning we give to all this!”[P].

#### 4.3.2. TPE Seen as a Fad

Some carers, in the context of the current strain on psychiatry units, considered that it was necessary to distance themselves from TPE by considering it to be an ephemeral

“trend” within the context of the current issues in psychiatry.[NA].

“There is a new thing, it’s a trend, it will go as it came, that’s it...”[NA].

### 4.4. MHPs’ Representations Could Hide a Lack of Knowledge or Awareness That Would Prevent Them from Appropriating TPE Programmes

#### 4.4.1. TPE Is Not Well Known

Some MHPs justified the opposition to TPE in psychiatry by a lack of knowledge of TPE among psychiatric carers. Others acknowledged that they did not have sufficient knowledge of TPE and had difficulty distinguishing it from psychoeducation, which is more widely practiced.

“There was nothing to stop me from advocating TPE: I just didn’t know about it!” …/….”For me, it wasn’t part of the care.”[P].

Psychoeducation programmes have been developed for people living with psychiatric disorders. They are defined as systematic didactic and psychotherapeutic interventions that aim to inform patients and their relatives about the psychiatric disorder and promote coping skills. Beyond the transmission of information, psychoeducation is a pedagogical method with the aims of identity clarification and empowerment [10]. The aim of TPE is to help patients acquire or maintain the skills they need to manage their lives with a chronic illness to the best of their ability, with the aim of acquiring and maintaining self-care skills and mobilising or acquiring coping skills based on the patient’s previous experience. The distinction between psychoeducation and TPE remains blurred.

“I talk about TPE, and in fact it’s systematic: every time, when people manage to understand a little bit what TPE is, they say, ’But in fact, we do it every day!’ Well, yes! Except that the representation that people have of TPE is not the right one! It’s not just about giving information to patients, to make them overly responsible and then let them manage.”[P].

“The training in TPE proved to me that people were like me, ignorant. Because in fact, we do a workshop... without a goal, without an objective, without a care project with psychoeducation! Whereas in TPE, there is a care project. We know what it should lead to and we know where the patient is at.”[NA].

Our study revealed a paradoxical situation in that although MHPs have been trained to provide TPE, they have difficulty distinguishing it from psychoeducation. It is possible that the quality of the training received needs to be reviewed.

#### 4.4.2. The Highly Formatted Framework of TPE Is Not Adapted to the Problems of Psychiatry

In France, TPE is considered as part of the patient care pathway. It aims to make patients more autonomous by facilitating their adherence to prescribed treatments and improving their quality of life. These programmes must be authorised by the Regional Health Agency (ARS). The conditions of authorisation provide a strict framework for the composition of the educational teams, the skills required to provide TPE and the teams that take the courses.

“We had our ARS file turned down many times; it was never right. The doctor persevered, we never got a positive response, we ended up giving up, and so each year we tried to modify the programme, to adapt it according to.”[NM].

“So, yes, honestly, it was all Greek to me. For me, it was really constraints that served no purpose; I didn’t see the point of submitting a file to the ARS.”[NA].

“Because it was totally beyond me, I didn’t really care” …/… “there are so many criteria... “…/… Doctor XXX, I think she must have made 4 or 5 reminders, files, and at the last one we said, is it really useful to continue, and we said no. So, we are not called a ‘Therapeutic Education Group’, we are called a ‘Psychoeducation Group’, we are not recognised, we are not registered in the official booklet of existing groups that we will see on the ARS.”[NA].

The conditions required for authorisation are poorly adapted to the functioning of psychiatric professionals, and have discouraged a number of initiatives. Although mental disorders are chronic illnesses, the TPE framework seems more suited to the management of a physical disorder because it was originally designed on a regulatory basis within that framework. Adaptations seem necessary.

#### 4.4.3. Mistaken Representations of TPE

Some caregivers reported that there was a form of refusal of the principle of TPE for some MHPs. This negative outlook was sometimes linked to a generational effect, explaining divergent representations of TPE.

“I think that doctors, even perhaps doctors or even nurses, like to control, to know everything, and not to allow too much autonomy, but, well...”[NA and SW].

“There was one major obstacle: a general practitioner who was totally opposed to all this.../.... Radically. So, it was impossible, even for things that are very common, namely, insulin injections, self-injections of insulin, it was impossible! You couldn’t... and that’s how it was!”[NA].

“Afterwards, it is perhaps also a question of generation but also undoubtedly of personality...” [P and Pharmacist].

#### 4.4.4. Lack of TPE Training in Initial/Continuing Education

Insufficient training in TPE, whether in initial and/or continuing education, has been identified as a major barrier to the dissemination of TPE knowledge and culture among psychiatric workers.

“So no, it was not at all a transversal thing in my training...”.../...“Yes, it was something optional, but it was optional, or rather over-optional.”[NA].

“It changed for the nurses, it was a portfolio to fill in with boxes, crosses and things, there were no more notes, so I was in my second year, so we saw, vaguely, therapeutic education in class, it must have been on the syllabus but really minor, eh: 2 h.” [NA].

## 5. Discussion

While the analysis of the various interviews we conducted with MHPs revealed many barriers, it also highlighted the means of improving the implementation of TPE for people with severe mental illnesses such as schizophrenia.

The conflict of values that we have brought to light shows that paternalism seems to be more widespread in psychiatry than what is generally assumed. There is indeed a long tradition of paternalism in psychiatry, even though patient rights have been considerably more in focus in recent decades [31]. From the point of view of the interviewees, patients with psychiatric disorders simply do not seem to be able to develop a sufficient level of self-control. It is on the basis of these arguments that the paternalistic approach would be justified in psychiatry. In this context, psychiatric staff members should be aware of their responsibility and not exploit dependency by making offers that would coerce or manipulate the patient’s free will [32,33]. One of the reasons cited for not moving towards more patient autonomy is the continuing reduction in the number of psychiatric beds [34]. This means that inpatients tend to be in more serious condition than in the past, and the proportion of those who are treated by restraint has increased.

One way to move towards a less paternalistic model could be to see psychiatry as a value-based practice [35] that would aim to (1) always start from the patient’s perspective but also seek a balance between legitimately different perspectives and (2) ensure that communication skills play an important role in clinical decision making. Although it is not always possible to move towards this mode of decision making in psychiatry, staff should at all times try to establish an open dialogue in order to reach a compromise that is acceptable to the patient and sufficiently appropriate from a professional point of view. Only after such a strategy has failed should staff consider paternalistic decision making [36]. In this sense, TPEs could play an interesting role in the paradigm shifts.

There also appears to be a lack of knowledge about TPE. In psychiatry, the educational dimension has been promoted for some years in effective, internationally recognised experiments, which are essentially based on the notion of empowerment and recovery [37]. The American Psychological Association states that sustainable recovery (from mental illness) requires treatment that is comprehensive, coordinated, consistent, competent, empathetic and person-centred [38]. Therapeutic education offered in an inappropriate manner may be ineffective, if not counterproductive [39]. Despite the obstacles, psychiatric carers trained in therapeutic education validate the fact that it is efficient in care and beneficial to their practice; psychiatrists are thus increasingly taking an educational approach [4].

TPE has shown its value in developing positive attitudes towards mental illness and mentally ill people and has led to improvements in self-reported health [39]. However, in France, Cadiot et al. [4] showed that very few structured programs were underway, and that most dealt with bipolar disorders and schizophrenia, i.e., disorders for which the effectiveness of patient education has been demonstrated. The French National Institute for Prevention and Health Education (INPES) estimates that only 50% of initial training facilities for health professionals offer specific teaching in TPE [40]. Limited access to training due to a lack of specific financial resources and overcrowded curricula is also a reason given by professionals [41], even though the legislation provides, for example, in the decree of 31 July 2009, for 150 h of training on preventive care for students of the State Nursing Diploma [42].

It would seem that TPE care is viewed as creating additional work for caregivers, causing a fear of insufficient return on investment. However, TPE should be a priority in the care plan since an early return home has become one of the key aims for supporting recovery from mental illness [43]. Mental health professionals have a scientific, ethical and moral responsibility to guide the social, political and health care organisations involved in the process of meeting mental health needs.

## 6. Recommendations and Limitations

This study has some limitations inherent to any qualitative study, such as possible self-selection bias or potential response bias. Due to the COVID-19 pandemic, some interviews were conducted by video conference (n = 2) or by telephone (n = 5). Telephone calls are impersonal compared to a physical interview and do not allow for reading, for example, body language or facial expressions. Although video conferencing is an interesting alternative to the telephone, it requires a certain amount of time to adapt to the tool so that the interviewer feels at ease, as does the interviewee.

Another limitation is that the views of those involved in psycho-education were not formally explored. In this context, it is difficult to say whether the concerns of the participants in the study are specific to TPE or related to general frustrations with the development of educational programmes in their institutions. This grey area of this study will be explored in the future.

We can nevertheless highlight some of the elements identified herein that could foster the ability of mental health professionals to deliver TPE to people with psychiatric disorders.

Continue the fundamental work on the recognition of the rights of mental health users.Continue efforts to destigmatise mental health, including among health professionals, and leave more place for the patient’s voice.Institutions must fully commit to accompanying the implementation of TPEs to achieve this, and it would be useful to organise a shared activity between professionals in order to create a dynamic for the implementation of TPE in psychiatry.Organisational or even financial support could be provided through shared resources between institutions, which could be organised through a periodical magazine, events, conferences or associations.Training for the provision of TPE should be fully integrated into initial training curricula and included in continuing education.Mental health nurses have a key role to play in TPE insofar as it is part of their skill set. In many European countries, important steps have been taken in recent years to recognise and develop the role of nurses in health care teams, particularly with regard to prevention, clinical monitoring and the education of chronic patients.The focus should be on the good practices needed to ensure equal access to care and on how to provide treatment at costs that are sustainable for health systems. In this case, ad hoc payments would promote the use of TPE.The framework for building TPE programmes and reporting to the ARS should be simplified so they can be easily recorded in the establishment’s care project.

## 7. Conclusions

Although TPE is of interest in the process of patient empowerment, we found that caregivers were reluctant to appropriate this approach to care. Efforts must be made in the initial and ongoing training of MHPs to move from a paternalistic model to a patient partnership model, which is made possible by TPE. These efforts must also be firmly supported by health care facilities, and proactive governance is required for the successful implementation of TPE.

## Figures and Tables

**Table 1 healthcare-10-01644-t001:** A comparison between therapeutic patient education or “training” and psychoeducation in France.

Educational Therapies	Therapeutic Patient Education or “Training”	Psychoeducation
Aims	Aims to encourage a process of empowerment in order to make the patient more autonomous and active in his/her behavioural changes throughout the care project.	Promotes the resolution of emotional, psychological, behavioural and cognitive problems
Disease type	For individuals with chronic diseases	For individuals with mental disorders
Legislative framework	Framed by the “Hospital, Patients, Health and Territory” law (HPST) of 21 July 2009 [10].	Does not require a legislative framework or compulsory training
Assessment	The “Haute Autorité de Santé” (HAS) has set recommendations for the annual and four-year self-assessment required for the renewal of programmes [11].	Evaluation procedures for the programmes developed are under the sole authority of the practitioners and the institutions that manage them.
Funding	Conditional funding by the “Agence Régionale de Santé (ARS)	No specific institutional funding

**Table 2 healthcare-10-01644-t002:** Population characteristics.

Demographics	n	(%)	Minimum	Maximum
Age (years) (SD)				
46.3 (8.2)	21	100%	34.2	59.6
**Gender**				
Male	4	19%		
Female	17	81%		
**Profession**			Cumulated frequency	Cumulative percentage
Nurse	8	38.1%	8	38.1%
Nurse Manager	4	19%	12	57.1%
Nursing assistant	2	9.5%	14	66.7%
Social worker	4	19%	18	85.7%
Pharmacist	1	4.8%	19	90.5%
Physician (1 doctor and 1 psychiatrist)	2	9.5%	21	100%
**Experience in psychiatry**				
0 to 5 years	3	14.3%	3	14.3%
6 to 10 years	5	28.2%	8	38.1%
11 to 15 years	2	9.5%	10	47.6%
>15 years	11	52.4%	21	100%

SD: Standard deviation.

## Data Availability

Data are fully available and will be shared upon request to C.R.

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
