# Peer review of "Caregiver Representations of Therapeutic Patient Education Programmes for People with Schizophrenia: A Qualitative Study"

_healthcare, 2022, doi:10.3390/healthcare10091644_

Round 1

Reviewer 1 Report

Thank you for the opportunity to review this paper. In light of the recovery movement in mental health this paper has significance and highlights the need for more education and the use of client involvement and empowerment in the delivery of care. In order to enhance the paper the following is suggested: 

1. Table 2: I am not sure this table adds to the paper and perhaps could be removed. It is sufficient to state that the participants came from 7 different organisations and were interviewed between August 2019 and December 2020.

2. Table 3: the heading needs to be with the table on page 208. Equally the heading of 'inclusion process and the beginning of this paragraph needs to be separated from Table 3.

3. Data Analysis heading should be placed over the page to page 6 instead of at the bottom of page 5

4. Results: Separate out the quotes and define what MHP they belong to. 

5. Funding: It is not clear if the funders had or had not a role in the study design data collections etc...perhaps this could be clarified.

Author Response

Reviewer 1:

Thank you for the opportunity to review this paper. In light of the recovery movement in mental health this paper has significance and highlights the need for more education and the use of client involvement and empowerment in the delivery of care. In order to enhance the paper the following is suggested:

Response:

Thank you

  1. Table 2: I am not sure this table adds to the paper and perhaps could be removed. It is sufficient to state that the participants came from 7 different organisations and were interviewed between August 2019 and December 2020.

Response:

We have made the requested change. Please see line 166 on page 5.

  1. Table 3: the heading needs to be with the table on page 208. Equally the heading of 'inclusion process and the beginning of this paragraph needs to be separated from Table 3.

Response:

We fully agree with your comment. Sometimes "bugs" can creep in during the submission process. We have corrected the pagination.

  1. Data Analysis heading should be placed over the page to page 6 instead of at the bottom of page 5

Response:

The change was made as requested.

  1. Results: Separate out the quotes and define what MHP they belong to.

Response:

Thank you for this suggestion. We have made the changes requested in the manuscript.

  1. Funding: It is not clear if the funders had or had not a role in the study design data collections etc...perhaps this could be clarified.

Response:

This research was funded by the French Ministry of Solidarity and Health, under a call for projects. As is the case with all calls for tender, which are very competitive, funders tend to look for answers to problems that raise questions in specific areas. Separate committees of funders ensure the selection of projects. We revised the “Funding source” section page 12 line 536 to be more precise.

Reviewer 2 Report

Summary of the key contribution of the paper:

Suggests that TPE needs large modifications in order to be useful in a mental health treatment plan

Highlights:

·        Table 1 is well constructed and helpful in understanding the differences between educational therapies

·        Page 4 lines 117-129 provide a good justification for the use of qualitative approaches in this study

·        Valuable point that MHPs have difficulty distinguishing TPE from psychoeducation (p. 8)

Lowlights:

·        On line 93 of page 3, the datapoint cited is from 2011. This is not reflective of the current landscape of TPE programs.

·        The paper should not be framed as a comparison of TPE and psychoeducation when no psychoeducation providers were interviewed

 The following criticisms are raised for the authors to address in their revision.

Q1:  The differences between training and psychoeducation are explained clearly, but not their similarities. Please add a graphic that shows the commonalities / shared characteristics of both treatments.

Q2: Specify who the “caregivers” are. Only physicians? Or also family members, etc. who take care of the patient?

Q3: On page 5 line 197, the authors specify that some interviews were collected in-person and some online. In the conclusions they should evaluate how this difference in modes may have influenced results.

Q4: Interview psychoeducation providers not trained in TPE and compare their responses to the previous interviewees. Right now cannot tell if the concerns of interviewees are specific to TPE or if they are general frustrations.

Typos:

i.                   Page 1 line 19 should be “the aim of this study”

ii.                 Page 3 line 97 should be “relationship”

Author Response

Reviewer 2:

Summary of the key contribution of the paper:

Suggests that TPE needs large modifications in order to be useful in a mental health treatment plan

Highlights:

  • Table 1 is well constructed and helpful in understanding the differences between educational therapies
  • Page 4 lines 117-129 provide a good justification for the use of qualitative approaches in this study
  • Valuable point that MHPs have difficulty distinguishing TPE from psychoeducation (p. 8)

Response:

Thank you for your valuable comments.

Lowlights:

  • On line 93 of page 3, the datapoint cited is from 2011. This is not reflective of the current landscape of TPE programs.

Response:

We have clarified this point lines 104 to 108 on page 3 and added a new reference.

[21]-Lang JP, Jurado N, Herdt C, Sauvanaud F, Lalanne Tongio L. [Education care in patients with psychiatric disorders in France: Psychoeducation or therapeutic patient education?]. Rev Epidemiol Sante Publique. 2019;67(1):59-64.

  • The paper should not be framed as a comparison of TPE and psychoeducation when no psychoeducation providers were interviewed

 The following criticisms are raised for the authors to address in their revision.

Response:

Thank you for the constructive comments which give us the opportunity to improve our manuscript.

Q1:  The differences between training and psychoeducation are explained clearly, but not their similarities. Please add a graphic that shows the commonalities / shared characteristics of both treatments.

Response:

We have added a new paragraph lines 82 to 86 page 3 to explain this point.

Q2: Specify who the “caregivers” are. Only physicians? Or also family members, etc. who take care of the patient?

Response:

A caregiver in mental health care refers to any person who is involved in the treatment or prevention of the illness or its complications. The doctor, nurse and care assistant can be considered caregivers. Please see lines 88 to 90, page 3.

Q3: On page 5 line 197, the authors specify that some interviews were collected in-person and some online. In the conclusions they should evaluate how this difference in modes may have influenced results.

Response:

Thank you for pointing out this limitation of the study which is related to the Covid crisis. We discussed this in the "Limitations" section lines 479 to 485 page 11.

Q4: Interview psychoeducation providers not trained in TPE and compare their responses to the previous interviewees. Right now cannot tell if the concerns of interviewees are specific to TPE or if they are general frustrations.

Response:

We agree, we have added a paragraph in the limitation section page 11 lines 486 to 490.

Typos:

  1. Page 1 line 19 should be “the aim of this study”

Response:

Thank you for point out this mistake. We made the correction.

  1. Page 3 line 97 should be “relationship”

Response:

Thank you for point out this mistake. We made the correction.

Round 2

Reviewer 2 Report

Summary of the key contribution of the paper:

Suggests that TPE needs large modifications in order to be useful in a mental health treatment plan

Highlights:

·        Table 1 is well constructed and helpful in understanding the differences between educational therapies

·        Page 4 lines 117-129 provide a good justification for the use of qualitative approaches in this study

·        Valuable point that MHPs have difficulty distinguishing TPE from psychoeducation (p. 8)

Lowlights:

Addressed all the comments.

The following criticisms are raised for the authors to address in their revision.

Q1:  The differences between training and psychoeducation are explained clearly, but not their similarities. Please add a graphic that shows the commonalities / shared characteristics of both treatments.

Addressed

Q2: Specify who the “caregivers” are. Only physicians? Or also family members, etc. who take care of the patient?

Addressed

Q3: On page 5 line 197, the authors specify that some interviews were collected in-person and some online. In the conclusions they should evaluate how this difference in modes may have influenced results.

Addressed

Q4: Interview psychoeducation providers not trained in TPE and compare their responses to the previous interviewees. Right now, cannot tell if the concerns of interviewees are specific to TPE or if they are general frustrations.

Addressed

Typos:

i.                   Page 1 line 19 should be “the aim of this study” Addressed

ii.                 Page 3 line 97 should be “relationship” Addressed
